# Artificial Intelligence-Based Microfluidic Platform for Detecting Contaminants in Water: A Review

**DOI:** 10.3390/s24134350

**Published:** 2024-07-04

**Authors:** Yihao Zhang, Jiaxuan Li, Yu Zhou, Xu Zhang, Xianhua Liu

**Affiliations:** School of Environmental Science and Engineering, Tianjin University, Tianjin 300354, China; zhangyihao_@tju.edu.cn (Y.Z.); ljxx@tju.edu.cn (J.L.); pidanzzy_22@tju.edu.cn (Y.Z.); zhangxu_2022@tju.edu.cn (X.Z.)

**Keywords:** artificial intelligence, microfluidic platforms, water pollutants, electrochemical detection, optical detection

## Abstract

Water pollution greatly impacts humans and ecosystems, so a series of policies have been enacted to control it. The first step in performing pollution control is to detect contaminants in the water. Various methods have been proposed for water quality testing, such as spectroscopy, chromatography, and electrochemical techniques. However, traditional testing methods require the utilization of laboratory equipment, which is large and not suitable for real-time testing in the field. Microfluidic devices can overcome the limitations of traditional testing instruments and have become an efficient and convenient tool for water quality analysis. At the same time, artificial intelligence is an ideal means of recognizing, classifying, and predicting data obtained from microfluidic systems. Microfluidic devices based on artificial intelligence and machine learning are being developed with great significance for the next generation of water quality monitoring systems. This review begins with a brief introduction to the algorithms involved in artificial intelligence and the materials used in the fabrication and detection techniques of microfluidic platforms. Then, the latest research development of combining the two for pollutant detection in water bodies, including heavy metals, pesticides, micro- and nanoplastics, and microalgae, is mainly introduced. Finally, the challenges encountered and the future directions of detection methods based on industrial intelligence and microfluidic chips are discussed.

## 1. Introduction

Water is the basis for the survival of all life. However, with the rapid development of industrialization and urbanization, pollution of the water environment has become one of the major problems facing the world [1]. Pollution of the water environment comes mainly from industrial wastewater, urban sewage, agricultural discharges, mineral extraction, and other activities [2]. Some of the pollutants commonly found in water bodies include inorganic pollutants, heavy metals, nutrients such as nitrogen and phosphorus, pesticides, and microplastics. The water environment can be harmed to varying degrees by these different types of pollution. Heavy metal pollution can lead to the death of aquatic organisms [3]. Pesticide pollution can be toxic to humans and animals [4]. Eutrophication pollution can cause algae blooms and deplete dissolved oxygen in the water, killing fish and other organisms [5]. Therefore, the detection of pollutants in water bodies can serve the purpose of early warning and avoid causing harm.

Traditional water quality monitoring methods include spectrophotometry, atomic absorption, mass spectrometry, and chromatography [6,7,8,9]. These methods face challenges such as long testing cycles, complexity, difficult sample handling, and the need for professionally trained personnel to operate them. Thus, there is a great need to establish technology that can detect pollutants in water in a simple, real-time, and rapid manner. Recently, great interest has been shown in combining microfluidics with artificial intelligence for water quality monitoring applications [10].

Microfluidics is a technology that enables liquids to be precisely manipulated and analyzed in micron or nanometer-scale space. Its basic feature and greatest advantage is that multiple unit technologies can be integrated at scale on an overall controllable micro-platform. It can control the flow and mixing of liquids to achieve a variety of reactions through the use of micromachining technologies such as microfluidic channels and microvalves. By precisely controlling the flow rate, microfluidics can realize a variety of functions such as chemical analysis, drug screening, and cell culture [11,12,13]. In addition, microfluidics can be designed as a multi-channel control system to enable parallel testing of multiple items. The samples to be tested are simultaneously divided into multiple reaction units, which are isolated from each other. It has the advantages of small sample size, high sensitivity, high integration, fast detection speed, and low cost. Overall, microfluidics has a wide range of application prospects due to its unique advantages in many fields such as biomedicine, drug discovery and development, and environmental monitoring [14,15,16,17].

Artificial intelligence (AI) is an emerging technological science that seeks to expand the theories, methods, techniques, and applied systems of human intelligence. It was developed to enable computer systems to perform tasks that require human intelligence, such as learning, reasoning, perceiving, and understanding language. In recent years, artificial intelligence has gained rapid development with the development of various neural networks and algorithms. It has already made great achievements in biomedicine, agriculture, material synthesis, and environmental monitoring [18,19,20,21]. Machine learning is an important branch of artificial intelligence that allows computer systems to learn from data and improve performance without explicit programming. Its algorithms include supervised, unsupervised, semi-supervised, and reinforcement learning. In addition, deep learning (DL) has become a popular direction in machine learning with the increasing amount of data and computational power [22]. Efficient processing of complex data is achieved by deep learning utilizing neural network structures to model the hierarchical structure of the human nervous system. Convolutional neural networks (CNNs) and Recurrent Neural Networks (RNNs) are two important branches of DL for image processing and sequence data modeling, respectively [23,24].

It is worth noting that while AI has great advantages, it also has some limitations. For example, the development of algorithms needs to be trained by collecting large amounts of data, and the lack of sufficient data may affect their performance. This is generally a very time-consuming and laborious task. The limitations of artificial intelligence can be overcome by microfluidic chip technology with the advantages of cost-effectiveness, large scale, and automation. The combination of the two allows for more efficient data acquisition and is a powerful tool for analyzing contaminants in water. Despite the increasing number of reviews on using microfluidics or artificial intelligence for detecting contaminants in water, there are few review papers that explore the combination of these two techniques for this purpose.

This review covers artificial intelligence-based microfluidic systems for water contamination detection. We begin with a brief overview of different microfluidic detection methods and AI algorithms. Then, a variety of contaminants are discussed in detail, including heavy metals, pesticides, micro- and nanoplastics, and microalgae, with a special focus on new detection strategies that utilize both AI and microfluidics. Finally, the prospects of both are summarized.

## 2. Artificial Intelligence

Artificial intelligence is a branch of the computer science discipline and is considered to be one of the three cutting-edge technologies of the twenty-first century. In the last three decades, it has enjoyed rapid growth and fruitful achievements in many disciplines. AI has gradually become an independent branch and has become a system of its own in both theory and practice.

Environmental pollution is one of the major issues of great concern in present-day society. Powerful technical support is provided to environmental protection and governance through artificial intelligence. The application of AI in the environment has covered several key areas, such as air quality monitoring, water quality monitoring and analysis, soil contamination identification, and sensor data collection [25]. Artificial intelligence-based methods have great advantages in environmental analysis because they can achieve real-time monitoring, accurate prediction, and intelligent decision-making by analyzing a large amount of data.

Machine learning (ML) is a multi-interdisciplinary discipline that specializes in how computers can simulate or achieve human learning behaviors to obtain new knowledge and skills. The performance of the self is constantly being improved through which existing knowledge structures are reorganized. There is a wide variety of AI algorithms, such as linear regression, logistic regression, decision trees, random forests (RF), support vector machines (SVM), K-nearest neighbor (KNN), neural networks, multilayer perceptron (MLP), and eXtreme gradient boosting (XGBoost) [26]. Each algorithm has its specific application scenarios and advantages. For example, linear regression algorithms attempt to find a best-fit straight line to predict future numerical outcomes; logistic regression is suitable for situations where the output is binary, such as classification problems. The main types of ML can be categorized as supervised, unsupervised, semi-supervised, and reinforcement learning. Supervised learning is training data that consists of known outcomes or labels, and the machine learning model adjusts by comparing its predictions to the actual results [27]. Unsupervised learning is training data without labels and the model needs to find out the intrinsic structure or patterns in the data [28]. Semi-supervised learning is where the training data is partially labeled and the model needs to learn both labeled and unlabeled data. Reinforcement learning maximizes some kind of reward signal by trying different behaviors in machine learning models that learn in interaction with the environment [29].

An artificial neural network (ANN) is an ML model derived from a network of neurons in the human brain, which consists of a hierarchical structure of multiple neurons or nodes. Each neuron receives inputs from the previous layer of neurons and passes the outputs to the next layer of neurons after processing with an activation function [30]. This structure allows complex nonlinear relationships to be learned by artificial neural networks. It overcomes the shortcomings of traditional artificial intelligence based on logical symbols in dealing with unstructured information and has the characteristics of self-adaptation, self-organization, and real-time learning.

Deep learning is a new machine learning method built on ANN. In deep learning, multilayer neural networks are utilized to learn a multi-level representation of the data, which allows for a better representation of the complex features and patterns of the data. More abstract and advanced feature representations are learned by deep learning models by increasing the depth of the network, which improves the performance and generalization ability of the model [31]. Multiple model architectures are involved in deep learning, each with its specific application areas and strengths [32,33,34]. The models often used as deep learning models include convolutional neural networks (CNNs), recurrent neural networks (RNNs), autoencoders, and deep reinforcement learning (DRL) (Figure 1). CNNs are designed to process data in the form of multiple arrays. They receive two-dimensional data structures and extract high-level features through convolutional layers. CNNs extract spatial correlations between neighboring data by computing the inner product of inputs and filters. The output is then pooled to reduce the spatial dimensions and generate high-level abstractions. RNNs transform input data into a form suitable for processing by the network and define how to recursively combine the data through recursive units. They use a loss function to measure the difference between predicted and true values and a back-propagation algorithm to compute the gradient in order to update the network parameters. After training is completed, the performance of the model is evaluated using validation and test sets to check for overfitting or underfitting and to make necessary adjustments. The first part of the autoencoder is the encoder, which is responsible for converting the input data into a compact intermediate representation. This process involves data downscaling and feature extraction. After the encoder, the data typically passes through a narrow layer called the bottleneck layer, which further compresses the data and forces the network to learn the most important features of the data. The decoder is the third part of the autoencoder that accepts the encoded data and tries to reconstruct the original input data. The training process of the autoencoder ensures that the reconstructed output is as close as possible to the original input. It finally uses a backpropagation algorithm to compute the gradient and an optimization algorithm to adjust the network weights to reduce the reconstruction error. DRL is a combination of deep learning and reinforcement learning. It requires, first, defining an environment suitable for the problem and then designing an intelligent body that can interact with the environment. The intelligence performs actions in the environment, and the environment responds to the intelligence’s actions and provides new states and rewards. It adjusts the parameters of its strategy or value network based on this feedback. Finally, DRL algorithms are used to optimize the strategy or value network of the intelligences.

Among these models, the CNN plays a vital role in environmental pollution monitoring or detection [36,37,38]. This is mainly due to its efficient feature extraction capability, its ability to adapt to different environments, and its real-time performance. First, key features in the input data (such as images) are automatically learned and extracted through a convolutional neural network that performs multilayer convolution and pooling operations. This ability to quickly and accurately recognize signs of environmental pollution provides strong support for subsequent monitoring and treatment. Second, it has a strong generalization ability and can adapt to input data in different environments, thus enabling accurate monitoring of the environment. Finally, it can process input data in real-time and respond quickly to environmental changes. However, this model may present a challenge in obtaining large-scale labeled data for training the model.

## 3. Microfluidic Platform Detection Technology

In microfluidic systems, liquids are typically flowed and mixed through microstructure devices such as microchannels, microvalves, and micropumps to enable efficient handling and analysis of trace samples [39].

### 3.1. Chip Substrate Material

A microfluidic chip is a miniaturized experimental platform, widely used in biomedicine, chemical analysis, environmental monitoring, and other fields in recent years [14,40]. The choice of its substrate material directly determines the performance and application range of the chip. Currently, common microfluidic chip substrate materials include silicon, quartz, glass, polymers, and paper.

#### 3.1.1. Silicon, Quartz, and Glass

Silicon was often used in microfluidic chip making in the beginning because of its good thermal stability and mechanical strength [41]. Silicon substrates are usually prepared using micromachining techniques such as etching and dicing. These methods can meet the requirements of microfluidic chips for high precision and high reliability by precisely controlling the geometry and size of the channels. However, silicon is limited in its range of applications due to its relatively poor biocompatibility and high cost. Glass and quartz, as traditional microfluidic chip substrate materials, have received widespread attention for their excellent optical properties, chemical stability, and high purity. Techniques such as machining and photolithography are commonly used to create glass and quartz substrate materials [42]. Their high transparency makes them ideal for optical inspection applications, while their chemical stability allows the chips to operate stably in a variety of chemical environments [43]. Li et al. [44] developed a portable cytometer based on a glass microfluidic chip with a more miniaturized size. Prepared by laser processing, the signal-to-noise ratio is greatly improved compared with polymer materials. It lays the foundation for miniaturized flow cytometry for clinical monitoring. However, they are somewhat limited in their widespread use due to the high cost of processing and the relative complexity of the process.

#### 3.1.2. Organic Polymer

Polymer materials are widely used in microfluidic chip fabrication due to their low cost, easy processing, and excellent biocompatibility. Common polymer materials used to make microfluidic chips include polymethylmethacrylate (PMMA), polydimethylsiloxane (PDMS), polycarbonate (PC), thermoplastic polyester (TPE), polystyrene (PS), and polyethylene glycol diacrylate (PEGDA) [45,46,47,48,49]. Polymer substrates are usually prepared using methods such as injection molding and hot pressing. These methods allow for mass production and lower chip costs, while the flexibility of the polymer material allows for a more flexible chip design. In addition, polymers can be further broadened by improving their wettability and biocompatibility through methods such as surface modification. However, they are relatively chemically unstable and may be eroded by certain chemicals, affecting the performance and life of the chip.

#### 3.1.3. Paper

Paper is a new microfluidic chip substrate material, which has attracted attention for its low cost, easy accessibility, and environmental friendliness [50,51,52]. Preparing paper-based microfluidic chips usually involves printing, cutting, and laminating techniques. Microchannels and chambers can be precisely fabricated on paper using inkjet printing or laser cutting techniques [53,54]. Multilayer structures and complex functions can also be realized by laminating different papers [55]. Paper-based microfluidic chips have a wide range of application prospects in bio-detection and chemical analysis and are especially suitable for rapid on-site detection and low-cost analysis. Wang et al. [56] presented a low-cost system for detecting hyperuricemia that combines a smartphone with a paper-based microfluidic chip. The detection limits for uric acid and creatinine were 0.0178 and 0.5983 mM, respectively, which were in high agreement with the hospital instrument. However, the relatively low strength and stability of paper-based materials may limit them in some applications where high precision and reliability are required.

### 3.2. Detection Method

High sensitivity and high throughput detection of trace samples are realized by combining the microfluidic system with various detection techniques. Current detection methods often used in conjunction with microfluidics include electrochemical detection, optical detection (fluorescence), electroluminescence (ECL) detection, colorimetric detection, mass spectrometry, and surface-enhanced Raman spectroscopy (SERS) detection.

#### 3.2.1. Electrochemical Detection Method

The electrochemical detection method is commonly used in microfluidics. It enables an electrical signal response to chemicals in solution by integrating microelectrodes on a microfluidic chip. This method has the advantages of high sensitivity, fast response time, and lower cost, and is particularly suitable for the detection of trace samples [57,58]. Siavash et al. [59] designed an aptamer-based electrochemical microfluidic biosensor to detect Cryptosporidium parvum. The feasibility of the adapted sensor was further demonstrated by detecting the targeted Haemophilus parvum in patient fecal samples. The assay results were consistent with microscopic examination and real-time quantitative polymerase chain reaction results.

#### 3.2.2. Optical Detection Method (Fluorescence)

Optical detection methods are another important method in microfluidics, with fluorescence detection being the most common. Quantitative and qualitative analysis of specific substances can be achieved by adding fluorescent markers to the sample and observing changes in the fluorescent signal using a micro-optical system in the microfluidic chip. They are characterized by high sensitivity and high selectivity and are particularly suitable for the detection of specific components in complex samples [60,61]. Lu et al. [62] designed a novel detection platform utilizing a fluorescent chip and a small analytical system for the immediate detection of sodium ions in human blood samples. The Na concentration was determined by the maximum fluorescence peak at 533 nm through the property that sodium green reagent reacts with Na ions to produce fluorescence. The correlation coefficient between the sodium ion concentration and the fluorescence intensity was 0.9956 and the limit of detection was 2.8 mM.

#### 3.2.3. Electroluminescence (ECL) Detection Method

The electroluminescence (ECL) detection method is a combination of electrochemical and optical properties. In a microfluidic chip, a luminescence phenomenon is generated by applying a voltage to excite the chemicals on the electrodes, and the luminescence signals are then detected and analyzed. The ECL method is particularly suitable for the rapid analysis of trace substances, due to the advantages of high sensitivity, low background interference, and real-time detection [63,64,65]. Zhu et al. [66] developed a microfluidic device utilizing electrochemiluminescence for the in situ detection of the heart-type fatty acid binding protein. The device can directly detect H-FABP in human serum without any pretreatment, with a linear range of 1–100 ng/mL and a low detection limit of 0.72 ng/mL.

#### 3.2.4. Colorimetric Detection Method

Colorimetric detection methods are based on the principle that a substance reacts in color with a specific reagent. In the microfluidic chip, the sample and reagents are mixed and reacted inside the chip by designing a miniature reaction chamber, and the presence or absence of specific components in the sample is determined by observing the color change after the reaction. Colorimetric methods are suitable for rapid on-site detection and initial screening because of their simplicity, low cost, and visualization [67,68]. Man et al. [69] proposed a colorimetric detection system for the quantitative detection of nitrite by combining a homemade colorimetric signal acquisition box with a smartphone app. The recoveries ranged from 94.92% to 105.60% by testing actual samples of lake, pure, and tap water.

#### 3.2.5. Mass Spectrometry Detection Method

Mass spectrometry is a method of separating and identifying compounds in a sample using a mass spectrometer. Efficient and accurate analysis of trace samples can be achieved by integrating a miniature mass spectrometer in a microfluidic chip or by coupling it with other mass spectrometry devices. It is characterized by high resolution, high sensitivity, and qualitative and quantitative features, and is particularly suitable for the identification and quantitative analysis of compounds in complex mixtures [70,71]. Xu et al. [72] presented a microfluidic chip-based capillary zone electrophoresis–mass spectrometry method for simultaneous measurement of ATP and related nucleotide analogs in biofluids. It can be used to determine airway inflammation and cough disorders in humans.

#### 3.2.6. Surface-Enhanced Raman Spectroscopy Detection Method

Surface-Enhanced Raman Spectroscopy (SERS) is a method that utilizes rough metal surfaces to enhance Raman scattering signals to achieve ultra-sensitive detection of molecules in samples [73]. By integrating a SERS substrate in a microfluidic chip, molecules in trace amounts of a sample solution can be adsorbed onto a rough metal surface, which can then be detected using Raman spectroscopy. It has very high sensitivity and specificity and is suitable for the identification and quantitative analysis of trace molecules. Huang et al. [74] developed a device combining surface-enhanced Raman and microfluidic chips for rapid antibiotic susceptibility testing. Bacterial incubation time is greatly reduced, and effective bacterial concentration is increased on microfluidic chips. This system was shown to enable highly sensitive and label-free bacterial detection to determine bacterial infection diseases.

In addition to the techniques we have mentioned, the use of microwave technology in combination with microfluidic devices for the detection of contaminants in water, such as heavy metals and microplastics, is also a very interesting area of research [75,76,77].

## 4. Application of AI-Based Microfluidic Platforms in Environmental Analysis

The coupling of microfluidic platforms and artificial intelligence networks has many advantages in water quality monitoring. The microfluidic platform can realize sample processing and analysis on a tiny scale. Combined with the high-speed computation and intelligent decision-making of an artificial intelligence network, it can realize the rapid detection and analysis of pollutants in water samples, which greatly shortens the detection time and improves the detection efficiency.

### 4.1. Heavy Metal Detection

Heavy metal pollution is the presence of heavy metal elements in the environment that exceed environmental quality standards or are harmful to living organisms. Heavy metals are a group of metallic elements that are denser and have larger atomic weights, such as Pb, Hg, Cd, Cr, and Ni [78,79]. They may come from industrial wastewater, agricultural discharges, municipal sewage, mine wastewater, and other sources. Water bodies and ecosystems are jeopardized by them, which may also hurt human health.

Commonly used methods for the detection of heavy metals include atomic absorption spectrometry (AAS), atomic fluorescence spectrometry (AFS), inductively coupled plasma mass spectrometry (ICP-MS), electrochemical methods, etc. Among them, the electrochemical methods are well suited to being combined with microfluidic devices for the detection of heavy metal ions in water [80].

Mercury is a common heavy metal found in soil and water throughout nature [81,82]. Mercury discharges from industrial wastewater may lead to the contamination of water bodies and mercury bio-enrichment in the environment, entering the food chain and causing harm to organisms in the food chain. It is very harmful to human health. Long-term mercury exposure may lead to neurological problems such as headaches, memory loss, and mood swings [83]. It may also lead to kidney damage and affect kidney function. Therefore, the detection of mercury in water bodies is essential. The combination of artificial intelligence and microfluidics can be a powerful tool for predicting heavy metal pollution. Pennacchio et al. [84] presented a portable biosensor that can detect mercury (II) in the nM range in seawater. The new chimera formed by the Vmh2 and H3w peptides adhered to a polystyrene multiwell plate, which could be measured by fluorescence analysis in conjunction with mercury (II). The detection limit was 0.4 nM in tap water and 0.3 nM in seawater. A machine learning-based approach to recognize sample fluorescence was developed. A dataset was compiled by extracting digital features from over 700 moving shots of 96 microtiter plates, which were then categorized and regressed by machine learning algorithms on concentration categories and actual fluorescence values. The developed biosensor was combined with three machine learning prediction models (multilayer perceptron, random forest, and XGBoost) to predict mercury concentrations without the use of traditional reader devices (Figure 2). This allows marine pollution to be monitored on-site by personnel without specialized training. Future research will aim to improve overall classification accuracy and reduce regression error.

### 4.2. Pesticide Detection

Pesticides are chemical substances or biological agents that control pests, weeds, fungi, and other pathogens. It plays an essential function in agricultural production and helps to improve crop yield and quality. However, improper use and management of pesticides can lead to environmental pollution, damage to ecosystems, and harm to human health. Detecting pesticides in water is one of the most important measures to protect water quality and human health [85,86,87]. Some of the commonly used methods and techniques to detect pesticides in water include chromatography, mass spectrometry, and spectroscopy. Meanwhile, the biosensor can be used to detect pesticide residues in water with high sensitivity and selectivity, which combines biological components and sensor technology [88,89].

Recently, with the rise of artificial intelligence, some commonly used detection techniques have been combined with machine learning to improve the efficiency of pesticide detection. Sahin et al. [90] presented a method to detect pesticides in water using the SERS technique and machine learning. Extracts of the plant cedar were utilized to generate silver nanoparticles directly on paper as a SERS platform, promoting the spontaneous flow of liquids with the help of the capillary forces of a paper-based microfluidic platform. The proposed method can detect four insecticides (myclobutanil, phosmet, thiram, and abamectin) while maintaining the advantages of environmental friendliness and low cost. Principal component analysis (PCA) was performed on a total of 583 Raman spectra of four pesticides to reduce the size of the data and to eliminate irrelevant or redundant data. Before running the algorithms, 385 Raman spectra were used as training data and the remaining 198 were used as test data. The principal components were then fed into different classification algorithms. Five different ML models were used for pesticide classification: linear kernel SVM, RBF kernel SVM, k-NN, DTs, and AdaBoost. The predictive accuracy of the k-NN method for each pesticide was 0.9641 for thiram, 0.9698 for myclobutanil, 0.9968 for abamectin, and 0.9997 for phosmet (Figure 3). In addition, the performance and classification accuracy of the proposed method can be improved by creating a database containing more pesticide types.

### 4.3. Micro- and Nanoplastic Detection

Microplastics, which are plastic particles less than 5 mm in diameter, are a major carrier of pollution [91]. They come in various shapes and are persistent, difficult to degrade, biotoxic, and bioaccumulative. They can be carriers of toxic metals, pesticides, and other pollutants and enter the body with air, water, food, and other carriers. A range of health problems can be triggered by long-term exposure to microplastics, including immune system disorders, neurological damage, and reduced fertility [92,93,94]. Microplastics come from a wide range of sources, including everyday household items and man-made plastic microbeads added to industrial products. In addition, plastic fragments with small particle sizes made from plastic waste broken by mechanical action and ultraviolet rays are also an important source of microplastics. Compared to microplastics, nanoplastics may be much more harmful to living organisms because they are more abundant and reactive. They have the potential to reach more remote locations and penetrate living cells [95,96]. The presence of microplastics in the environment and the various identified and unidentified hazards to organisms have received widespread attention.

Several methods have been used in microplastic detection and identification with some success: for example, microscopic observation, infrared spectroscopic analysis, Raman spectroscopic analysis, chromatography–mass spectrometry techniques, flow cytometry, and biosensing techniques [97,98,99,100]. These methods can be used individually or in combination to more fully analyze microplastic particles in environmental samples.

Recently, digital holography (DH) has been used to image MPs and has demonstrated its potential utility in the physical characterization of plastic-like particles [101,102]. Meantime, because of the rapid advancement of AI, models based on certain customizations help to enhance the performance of certain detection approaches [103]. Valentino et al. [104] proposed a method in which the ability of a digital holography (DH) system to recognize microplastics is increased by utilizing artificial intelligence. The microfluidic chip is designed to be single-channel, consisting of an inlet and outlet port that images the sample as it flows through the chip channel. The DH-wrapped phase diagrams of different species of diatoms and MPs were segmented, and the segmentation took into account two object supports, one for the shape of the specimen and the other for the roughness of the specimen. The generated binary masks are multiplied by the phase contrast maps for extracting several features, including texture correlation, roughness kurtosis, and more. Four classifiers were trained and tested using fractal features: a decision tree, a shallow convolutional neural network (CNN), a K-nearest neighbors (KNN), and a support vector machine (SVM). MPs and 55 classes of diatoms were classified using fractal features, of which the best classifier was the support vector machine with an accuracy of 98.5% (k = 5) (Figure 4A). The possibility to identify and monitor microplastics directly was ensured by utilizing a microfluidic chip platform for flow field acquisition of real marine samples. At present, this device can only be used to distinguish between MPs and diatoms. There is a lot of particulate matter in real environments, and further consideration needs to be given to how to differentiate between MPs and other particulate matter if it is to be truly applied in the field. Gong et al. [105] proposed a new microfluidic approach for capturing and identifying microplastics in surface seawater without labeling. A microfluidic device with a sieve-like structure specifically designed to capture small-sized particles was introduced. This device was made with PDMS by soft lithography, and a glass sheet was placed on top of the channel to form a closed channel with no permanent bonding. The cover can be removed to eliminate potential background signals from smaller particles. The researchers constructed a comprehensive training dataset by combining samples collected in the lab with publicly available datasets. Enhanced data is pooled with raw spectral data, and they have further optimized models. Various models including support vector machine (SVM), random forest (RF), convolutional neural network (CNN), and residual neural network (ResNet34) were investigated to evaluate their performance in recognizing a wide range of common plastics. The results of the study show that the CNN outperforms other models and can achieve an accuracy of 93% with an average area under the curve of 98 ± 0.02% (Figure 4B). In addition, field tests have demonstrated that the designed microfluidic device can effectively capture and identify microplastics smaller than 50 μm. Water samples collected in the field first need to be manually filtered, and this device needs to be transported from the field back to the laboratory for subsequent analysis, so true automation and on-site testing have not yet been achieved. Future research efforts should focus on continuously expanding the dataset by incorporating a wider range of environmental samples. In addition, it is crucial to improve deep learning models to increase accuracy and robustness.

### 4.4. Microalgae Detection

Microalgae are those tiny groups of algae whose forms can only be recognized under a microscope and are one of the plankton in the water column. They are a class of plants with a simple structure, tiny individuals that like to float freely with the seawater. Microalgae can generally photosynthesize, thus contributing to the orderly functioning of marine ecosystems [106]. However, some negative effects may also be brought about by it. For example, aquatic ecosystems and biological health are jeopardized by their overproliferation, in some cases, to form harmful algal blooms [107,108].

For the detection of microalgae, methods such as flow cytometry and chlorophyll autofluorescence are used for the assessment of their activity [109,110,111]. The activity of microalgae was judged by flow cytometry detection of the fluorescence intensity of cells, and the internal chlorophyll content and photosynthetic intensity of microalgae were inferred by chlorophyll autofluorescence. Traditional methods of detecting and identifying microalgae have relied on field sampling, with samples being transported to laboratories for analysis. This approach is time-consuming and costly, while the activity of microalgae may be affected during transportation.

The use of machine learning algorithms to assist in the analysis and detection of microalgae in the laboratory has long been reported [112,113]. However, an integrated platform approach that enables real-time and in situ analytics has yet to emerge. To address this issue, Göröcs et al. [114] designed an imaging flow cytometer that can be used for in situ detection of plankton, which supports deep learning. The device is based on partially coherent, lensless, holographic microscopy technology and is capable of acquiring diffraction patterns of flowing micro-objects within microfluidic channels (Figure 5A). A microfluidic single channel (Ibidi µ-Slide I) with an internal height of 0.8 mm was placed on top of the image sensor, secured using a three-dimensional (3D) printed bracket and connected to a peristaltic pump. These holographic diffraction patterns were re-established in real-time by using deep learning-based phase recovery and image reconstruction methods. Various phase recovery reconstructions of aqueous micro-objects captured by a device are pre-trained through the use of a convolutional neural network. This method can automatically and accurately acquire the spectral morphology of an object without the use of external markers, generating a color image of the micro-object. In addition, motion blur was eliminated by simultaneously illuminating the sample with pulsed red, green, and blue light-emitting diodes. The proposed device is less costly and smaller in size compared to flow cytometers in the laboratory. Concentrations of an alga (Pseudo-nitzschia) in six public beaches in Los Angeles were measured with this device and were in general agreement with measurements made by the Department of Public Health. Liao et al. [115] designed a portable microfluidic device that automatically collects and analyzes algae in wild waters. The device integrates a lensless algae image acquisition module, algae segmentation and classification circuitry, and a touch screen, featuring low cost and miniaturization. The image acquisition module is mainly composed of a microfluidic single-channel chip and a lensless, embedded CMOS image sensor. The module uses an LED light source and places the microfluidic channel chip on the CMOS light-sensitive surface for imaging to obtain the holographic image of the algal cells. The FPGA processing module first segments the collected images using a segmentation network, and then classifies and counts the segmented algae using a classification network. Meanwhile, to solve the problem of low image resolution of lensless imaging systems, two algorithms were designed with deep neural networks as the algorithmic framework: the algae segmentation algorithm and the algae classification algorithm. Dual asymmetric quantization algorithms and circuit structures for hardware implementation of deep neural networks are proposed with a device accuracy of 94.27% (Figure 5B). With ongoing research in AI algorithms, Zheng et al. [116] proposed an Automated Intelligent Microfluidic Platform (AIMP) that can detect and categorize four microalgae (Cosmarium, Closterium, Micrasterias, and *Haematococcus pluvialis*) through automated control and intelligent data analysis. The platform combines a low-cost portable USB microscope with a miniature motorized stage to miniaturize an otherwise expensive lab microscope while maintaining critical functionality. Each microfluidic chip has four sets of sample detection channels, each consisting of an inlet, an outlet, and an observation grid of nine identically sized observation chambers. The microfluidic chambers are fabricated by laser-cutting double-sided adhesive tape. One side of the tape is adhered to a glass surface and the other side is adhered to a transparent PMMA plate. Four different sizes and morphologies of microalgae were selected to create the dataset used to train the microalgae detection ML network. From the initial 812 captured images, 630, 770, 737, and 736 labels were extracted for Cosmarium, Closterium, *Haematococcus pluvialis*, and Micrasterias, respectively, to create the raw image dataset. A dataset of these microalgae was collected and the microalgae species detection network (MSDN, based on the YOLOv5 architecture) trained on the established dataset achieved an accuracy of 92.8% at 0.5 intersection-over-union (mAP@0.5) (Figure 5C). The performance of the device is currently limited by a relatively small dataset, but it will get better as more high-quality data is integrated. Detailed information on the performance of the various devices is given in Table 1.

## 5. Conclusions and Future Perspective

Water pollution is a serious problem in the world nowadays. It can not only have a serious impact on human health but also cause damage to the ecosystem. That is why it is essential to detect and monitor pollutants in water bodies. Microfluidic platforms offer significant advantages over traditional detection methods in real-time, on-site, and rapid detection, but there are some shortcomings in data analysis. Artificial intelligence technology may be a promising candidate in this respect. Since the microfluidic chip requires only a very small amount of samples to generate data, AI can respond more quickly and improve the reliability of monitoring based on the large amount of data generated by the microfluidic chip. The combination of microfluidics and artificial intelligence techniques has been shown to be a prospective tool for analyzing contaminants in water bodies. Currently, there are already some commercialized products monitoring water quality based on microfluidic chip technology and artificial intelligence. For example, some water quality detectors were designed based on microfluidic microspectroscopy technology to detect three important indicators of water quality including total organic carbon (TOC), chemical oxygen demand (COD), and turbidity. The equipment relies on spectral technology to collect the spectra of substances in the water and utilizes AI intelligent algorithms to determine specific substances to achieve the effect of testing water quality.

Although the advantages of combining AI with microfluidic chips for water quality analysis have been confirmed by this review, some potential challenges remain. First, although AI algorithms show great advantages in data analysis, they still need to be continuously optimized to ensure their accuracy and reliability when dealing with the complex data generated by microfluidic chips. Second, a large amount of high-quality training data is needed by efficient AI models. However, obtaining sufficiently large amounts of labeled data can be a challenge in real-world water quality analysis. The performance of the model is affected by the lack of data and the difference in data quality. Finally, although microfluidic chips and AI technologies have great advantages in theory, cost is still an important factor to be considered in practical applications. Especially in developing countries with limited resources, an important challenge is how to reduce the cost of the technology and make it more widely available for water quality monitoring.

As technology continues to evolve, future research directions include: (1) development of smarter algorithms: further optimize and develop artificial intelligence algorithms applicable to microfluidic chip data analysis to enhance the accuracy of contaminant detection; (2) device-integrated design: develop portable devices that integrate microfluidic chips and artificial intelligence data processing units to achieve rapid on-site detection and analysis; (3) multi-parameter detection: simultaneous detection of multiple pollutants in water bodies needs to be realized to improve the comprehensiveness and accuracy of environmental monitoring.

## Figures and Tables

**Figure 1 sensors-24-04350-f001:**
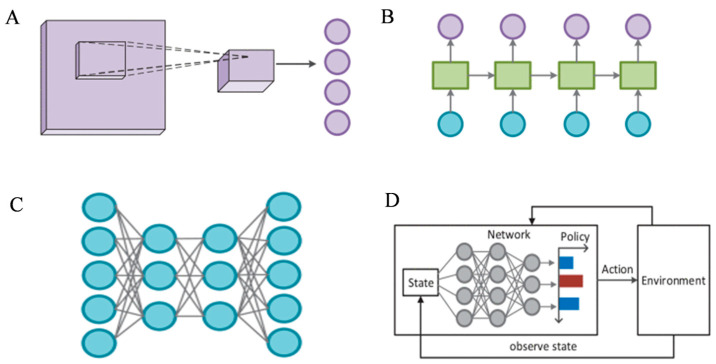
The structure of deep learning models. (**A**) convolution neural network. (**B**) recurrent neural network. (**C**) autoencoder. (**D**) deep reinforcement learning. Reprinted with permission from ref. [35]. Copyright 2020 IEEE.

**Figure 2 sensors-24-04350-f002:**
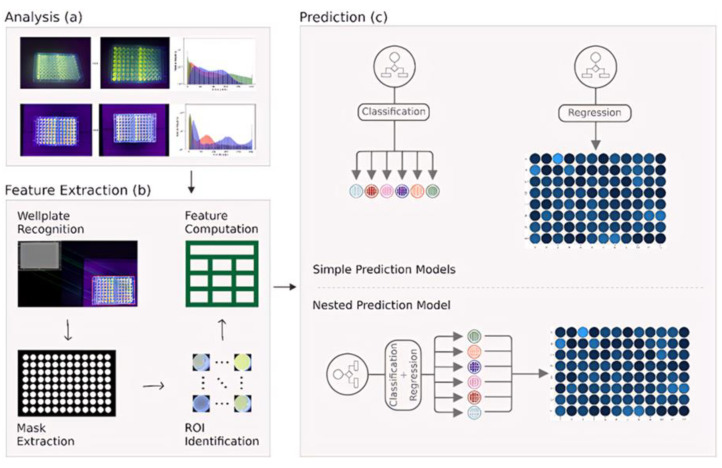
Schematic diagram of the workflow of the detection device and its stages. (**a**)Preliminary analysis of images acquired under different lighting conditions. (**b**) A reference image is selected and the area of the well plate on the photograph is identified. A mask is then generated for each image and the image containing each well, i.e., the ROI, is extracted. from this, the numerical features associated with each sample are calculated and stored in tabular form. (**c**) The obtained dataset was fed into the ML model to classify and regress the concentration categories and actual fluorescence values, respectively. As well as a nested model designed. Reprinted with permission from ref. [84]. Copyright 2022 Elsevier.

**Figure 3 sensors-24-04350-f003:**
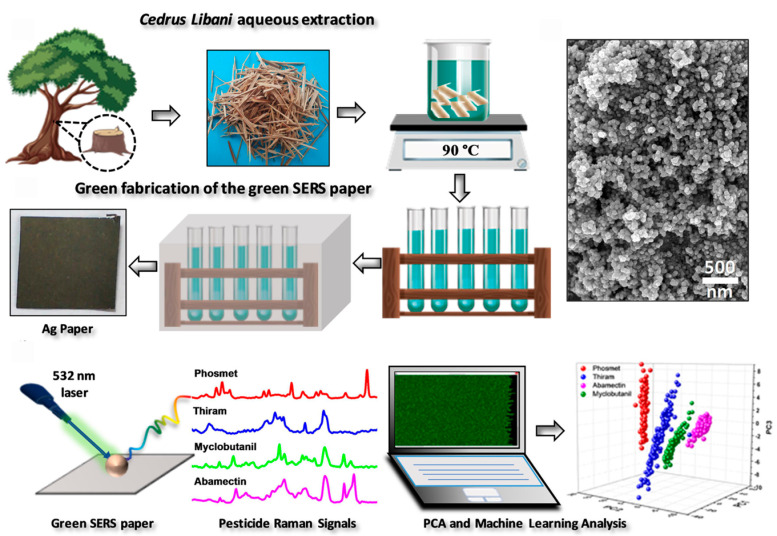
Schematic diagram of the method for preparing green SERS paper chips. Reprinted with permission from ref. [90]. Copyright 2022 American Chemical Society.

**Figure 4 sensors-24-04350-f004:**
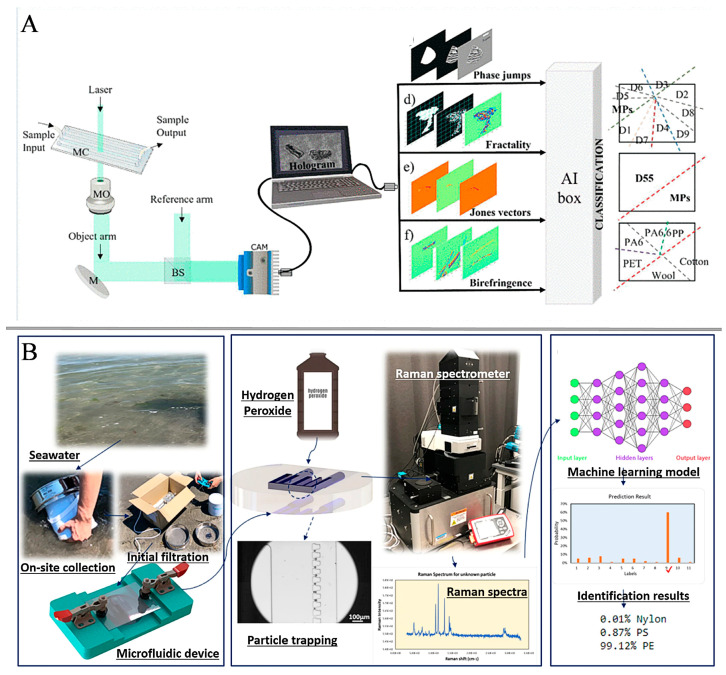
(**A**) Schematic diagram of DH-AI detection device. Reprinted with permission from ref. [104]. Copyright 2022 IEEE. (**B**) Simplified schematic of the process of identifying marine microplastics. Reprinted with permission from ref. [105]. Copyright 2023 Springer Nature Limited.

**Figure 5 sensors-24-04350-f005:**
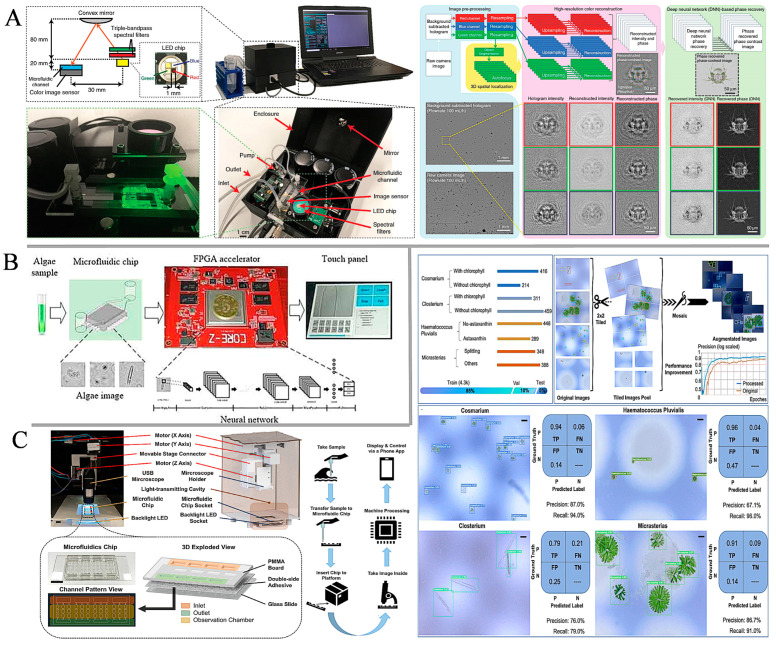
(**A**) Imaging photographs of flow cytometry devices and algorithms for object segmentation and deep learning-based hologram reconstruction. Reprinted with permission from ref. [114]. Copyright 2018 Springer Nature Limited. (**B**) Image of portable microfluidic lensless deep neural network algae monitoring device. Reprinted with permission from ref. [115]. Copyright 2021 IEEE. (**C**) Schematic of AIMP and plots of dataset creation and target detection performance for four microalgae. Reprinted with permission from ref. [116]. Copyright 2024 Royal Society of Chemistry.

**Table 1 sensors-24-04350-t001:** Detection of contaminants in water using artificially intelligent microfluidic system.

Pollutant		Chip Materials	Detection Technology	AI	Accuracy	Limit of Detection	Ref.
heavy metal	Hg^2+^	polystyrene	Optical detection method (fluorescence)	Multilayer Perceptron, Random Forest, and XGBoost	95%	0.4 nM in tap water and 0.3 nM in sea water	[84]
pesticides	abamectin, myclobutanil, phosmet, thiram	paper	Surface-Enhanced Raman Spectroscopy	k-nearest neighbors (k-NNs)	92.46%	1 μM	[90]
Micro- and nanoplastics	polyethylene, polystyrene, polyethylene terephthalate, polypropylene, polyvinyl chloride		DH	support vector machine (SVM)	99.3%		[104]
	polystyrene, polypropylene, polyethylene, polyamide, polyester, polyethylene terephthalate, polyvinyl chloride, polyurethane, polycarbonate, poly(methyl methacrylate), and cellulose acetate	PDMS	Raman spectroscopy	convolutional neural networks (CNN),	93%		[105]
microalgae	Ceratium furca, Pseudo-nitzschia	Polymers	flow cytometer	convolutional neural network			[114]
	anabaena, quadricauda, closterium		lensless imaging	deep neural network algorithm	94.27%		[115]
	Cosmarium, Closterium, Micrasterias, *Haematococcus pluvialis*	PMMA\Glass	USB microscope	MSDN (based on the YOLOv5 architecture)	92.8%		[116]

## Data Availability

The data used to support the findings of this study are available from the corresponding author upon reasonable request.

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
