# Peer review of "Artificial Intelligence-Based Microfluidic Platform for Detecting Contaminants in Water: A Review"

_sensors, 2024, doi:10.3390/s24134350_

Round 1

Reviewer 1 Report

Comments and Suggestions for Authors

The review, " Artificial intelligence-based microfluidic platform for detecting contaminants in water: a mini review " presents a good overview of the recent achievements in microfluidic platform for detecting water contaminates using artificial intelligence. The paper is well-written and covering a very important area. I've the following comments:

1.     In the introduction part, the authors clearly outline the significance of detecting contaminants in water due to their impact on human and ecological health as well as the use of microfluidics and AI technology. However, the authors could further clarify the unique contributions of their review in the context of existing literature.

2. You may consider including recent work using microwave technology in microfluidic platforms for water contaminates detection. https://doi.org/10.1016/j.sna.2023.114652; 10.1109/IMS37962.2022.9865376https://doi.org/10.3390/s19102393.

3.     Given the emphasis on microfluidics, it would be beneficial to include microfluidic devices in the table along with the Limit of Detection for the sensing results. This addition would provide a clearer and more comprehensive overview of the methods in the table.

4.     Since this review is focusing on AI and microfluidics, section 4 would greatly benefit from more in-depth technical details. Detailed description of the microfluidic designs and their integration with AI components would enhance the reader's understanding of how these technologies converge and operate together effectively.

5.     Are there any real-world applications that have successfully integrated AI with microfluidics for water quality monitoring? If so, please include.

Comments on the Quality of English Language

The review, " Artificial intelligence-based microfluidic platform for detecting contaminants in water: a mini review " presents a good overview of the recent achievements in microfluidic platform for detecting water contaminates using artificial intelligence. The paper is well-written and covering a very important area. I've the following comments:

1.     In the introduction part, the authors clearly outline the significance of detecting contaminants in water due to their impact on human and ecological health as well as the use of microfluidics and AI technology. However, the authors could further clarify the unique contributions of their review in the context of existing literature.

2.     You may consider including recent work using microwave technology in microfluidic platforms for water contaminates detection. https://doi.org/10.1016/j.sna.2023.114652; 10.1109/IMS37962.2022.9865376; https://doi.org/10.1016/j.sna.2022.113733

3.     Given the emphasis on Microfluidics, it would be beneficial to include microfluidic devices in the tables along with the limits of detection for the sensing results. This addition would provide a clearer and more comprehensive overview of the capabilities and performance metrics of these devices in various detection scenarios.

4.     Since this review is focusing on AI and microfluidics, section 4 would greatly benefit from more in-depth technical details. A thorough description of the microfluidic designs and their integration with AI components would enhance the reader's understanding of how these technologies converge and operate together effectively.

5.     Are there any real-world applications that have successfully integrated AI with microfluidics for water quality monitoring? If so, please include.

Reviewer 2 Report

Comments and Suggestions for Authors

1.In the second chapter of the article, the author provides less introduction. Specific operation modes of various deep learning models need to be added.

2.In section 4, there is a lack of necessary analysis of specific applications of AI. For example, section 4.1 does not describe the way in which biosensors are combined with the three machine learning predictive models.

3.In Chapter 5, the author elaborates on the shortcomings and further development directions of combining artificial intelligence and microfluidic chips for detection. However, these shortcomings and further development directions were not pointed out in Chapter 4. The content of Chapter 4 should be supplemented and corrected.

Reviewer 3 Report

Comments and Suggestions for Authors

The paper is a mini review about the usage of AI for detecting contaminants in water using microfluidic systems.

The author introduces the basic concepts of AI and microfluidics and shows seven systens that uses Ai and microfluidics.

The paper is a very good mini review.

The only thing that was not clear to me is the usage of the word "image" in line 150,154 and 155. I could not make the link between the environmental pollution monitoring and the word image. The data for pollution monitoring are not only images. It would be nice to clarify this. 
